# Potential Maneuvers for Providing Optimal Tidal Volume Using the One-Handed EC Technique

**DOI:** 10.3390/healthcare10081365

**Published:** 2022-07-23

**Authors:** Dongchoon Uhm, Ajung Kim

**Affiliations:** 1Department of Emergency Medical Technology, Daejeon University, 62 Daehak-ro, Dong-gu, Daejeon 300-716, Korea; dchuhm@dju.kr; 2Department of Emergency Medical Technology, Kyungil University, 50 Gamasil-gil, Hayang-eup 38428, Korea

**Keywords:** bag-valve-mask, tidal volume, EC technique

## Abstract

Bag-valve-mask is a device that manually provides positive oxygen pressure. The grip technique of the character E and C shape is recommended to carry out this effectively. However, when applying this method, the hand in which the direction of the EC technique should be performed and the degree of adhesion pressure while performing the technique are unknown. This study aims to identify the factors influencing tidal volume and to determine the ideal sealing method between mask and face in the one-handed EC technique to optimize the Vt. A simulation study was conducted using a mechanical lung model in a scenario that resembled respiratory arrest. Multiple regression analyses identified high peak pressure, high left spot adhesion strength of the mask, and low right spot and bottom spot adhesion strength of the mask as significant factors. To provide an optimal Vt, it may be necessary to apply more strength to the left area of the mask when forming the “C” shape with the thumb and index finger of left hand using the one-handed EC technique.

## 1. Introduction

The most effective method of oxygenation in patients with respiratory failure or cardiac arrest (CA) is the use of bag-valve-mask (BVM) ventilation. BVM ventilation is recommended over endotracheal tube intubation for prehospital CA [1], although there is no difference between these methods in the return of spontaneous circulation [2].

Adequate BVM ventilation is strongly related to the survival rate and oxygenation of patients with cardiac and respiratory arrest. Excessive ventilation reduces survival rates since the intrathoracic pressure is increased and the coronary perfusion pressure is decreased [3]; however, due to hypoxia, there is a decline in the survival rate with the inadequate ventilation [4,5,6]. In addition, a peak pressure (PP) of ≥ 40 cmH_2_O results in a diminished survival rate due to oxygen toxicity, lung damage, hemodynamic problems, and neuromuscular complications [7,8]. Therefore, maintaining optimal ventilation without exceeding the maximum PP is crucial. 

Previous studies have reported that individual characteristics such as sex, weight, height, mask grip method [9,10,11], confidence [12], hand size, width, strength, and dominance [9,10,11] affect tidal volume (Vt). These can be divided into improvable and unimprovable parameters to provide adequate Vt for patients in emergency situations. Although the patients’ physical characteristics are unchangeable, various studies [12,13] have attempted to find a method for providing an appropriate Vt using BVM ventilation. However, regardless of the method used, hyperventilation cannot be avoided. 

Traditionally, ventilation with approximately 500 mL of air to produce a visible chest rise using the one-handed EC technique has been recommended. This technique involves complete mask sealing against the face with the thumb and index finger of one hand wrapped in a “C” shape around the mask apex and the remaining fingers in an “E” shape, while lifting the jaw for effective BVM ventilation. Successful ventilation via the EC technique can be achieved by creating a tight seal between the mask and face and squeezing the bag with reasonable force [14]. If a full seal is not achieved, air leakage occurs, which can be assessed by measuring the adhesion strength between the contact surface of the mask and the patient’s face. However, it has not yet been reported which contact surface should be sealed and with what force or how to position the hand during the process to provide adequate Vt. Identifying the factors affecting Vt during the one-handed EC grip will be helpful in providing adequate Vt. 

Therefore, this study aims to determine the most suitable sealing method of the hand between the mask and face for providing optimal Vt while using BVM ventilation with the one-handed EC technique. 

## 2. Materials and Methods

### 2.1. Study Design

This simulation study aimed to determine the optimal technique for providing the appropriate Vt using the one-handed EC grip in an adult respiratory arrest scenario with a mechanical lung model. 

### 2.2. Participants and Setting

To yield an adequate amount of sample data, the G*Power version 3.1.2. (Heinrich Heine University, Düsseldorf, Germany) was used. A test power (1−β) of 0.95, an effect size of 0.15, and an alpha of 0.05 for 15 predictors were required for 74 datapoints; in this study, one datapoint referred to one ventilation. In respiratory assistance therapy, ventilation is performed once approximately every 5–6 s; therefore, 20–24 ventilations are performed in 2 min. Thus, four participants were required (20~24 datapoints/2 min × 4 participants = 80~96 datapoints). 

Although, to the data reliability, the study aimed to enroll a total of 33 participants (20~24 datapoints/2 min × 33 participants = 660~792 datapoints), 91 undergraduate paramedic students who had completed a course related to professional airway management via convenience sampling at Daejeon University, South Korea, were enrolled. Since two students dropped out for personal reasons, 89 participants were finally included in the analyses (Table 1).

### 2.3. Ethical Consideration and Data Collection

This study was approved by the relevant institutional review committee (1040647-202002-HR-007-02) and upheld the principles of the Declaration of Helsinki. The researchers contacted the Dean of the Department of Emergency Medical Services at the D University to obtain permission for recruitment. Participation was voluntary and anonymous; written informed consent was obtained from all participants before the commencement of the study. The participants were informed that they could withdraw their consent at any point during the study without any consequences. The data were collected between 15 October and 31 October 2020. 

### 2.4. Experimental Setting and Variables

In this study, the general characteristics (gender, grade, mask sealing hand) and PP were set as control variables and 4-spot (mask apex, bottom, left, and right) adhesion strength between the mask and the mannequin face as independent variables, with Vt as the dependent variable. The general characteristics were collected the day before the simulation study. The experimental variables, such as Vt, PP, and 4-spot adhesion strengths, mask-sealing hand were collected by a trained research assistant. Before collecting the experimental variables for each participant, the trained research assistant placed the BVM in the same position in the manikin to prevent errors or bias. The data were collected in a scenario of a patient with respiratory arrest, by performing ventilation for 2 min on the mechanical lung model without practice. 

### 2.5. Tidal Volume and Peak Pressure

The RespiTrainer^®^ Advance manikin (Quick Lung Advance, 2008; IngMar Medical, Ltd., Pittsburgh, PA, USA) was used. It was positioned at a height midline to the participants’ femur to minimize fatigue during ventilation [15,16]. The Quick Lung^®^ and personal digital assistance (PDA) devices were connected to the RespiTrainer^®^ Advance manikin. The data such as Vt and PP were delivered to the PDA via Bluetooth and calculated automatically. The Quick Lung^®^ was set at a compliance of 50 mL/cmH_2_O and a resistance of 5 cmH_2_O/L⋅s according to the methods used in previous studies [17,18].

All the participants were asked to use a BVM ventilator (1600 mL, Ambu Mark IV—Reusable Resuscitator, with size 5 silicon face mask; Ambu, Copenhagen, Denmark) to ventilate the manikin in simulated respiratory arrest for 2 min. The ventilation was performed with the participants in a standing position at the head of the manikin. 

### 2.6. Four-Spot Adhesion Strength

To measure the 4-spot adhesion strength between the mask and the manikin’s face using the EC technique, sensors were attached to the four contact parts (apex, left, right, and bottom) that the mask touched when it was placed on the face of the manikin (Figure 1). Four 0.5” circles (Part No. 402) of the Interlink Force Sensing Resistors (Interlink FSR^®^: 148 Interlink Electronics, Irvine, CA, USA) measured the adhesion strength. This sensor is a pressure sensor. To obtain the value recognized by the sensor, the sensor was connected to a specially made ARDUINO kit. The data were coded and calculated using the ARDUINO software.

The selection of the mask-sealing hand (the allocation of hands to hold the bag and mask of the BVM ventilator) was decided upon by each participant. Thereafter, the research assistant checked the hand in the direction used by the subject, visually, before the data were collected.

### 2.7. Data Analyses

Data were analyzed using SPSS Statistics for Windows, Version 27.0 (IBM, Armonk, NY, USA). Logarithmic and squared transformations were used for the variables with kurtosis and skewness that were below the standard of normality (skewness < 3, kurtosis < 10) [19]. The statistical analyses included the use of descriptive statistics, Pearson’s correlation coefficient, and a multiple regression analysis. In the multiple regression analysis, the control variables (sex, school year, mask-sealing hand, and PP), independent variables (4-spot adhesion strength), and dependent variables (Vt) were used. In addition, multicollinearity was defined as a variation inflation factor ≤5. The statistical significance was set at *p* < 0.05. 

## 3. Results

### 3.1. General Characteristics

The participants comprised 50.1% males and 49.9% females; 34.1% were sophomore students, 36.2% juniors, and 29.7% seniors. Most of the participants (92.6%) held the mask in their left hand. The mean Vt and PP were 415.98 ± 102.4 mL and 15.56 ± 5.62 cmH_2_O, respectively. The mean 4-spot adhesion strengths at the apex, bottom, right, and left of the mask were 0.03 ± 0.07, 0.69 ± 0.63, 0.62 ± 0.49, and 0.17 ± 0.29 N, respectively (Table 2).

### 3.2. Correlation between Vt, PP, and 4-Spot Adhesion Strengths

The Vt showed a positive correlation with the PP (*r* = 0.744, *p* < 0.001), apex (*r* = 0.083, *p* = 0.001), and left (*r* = 0.280, *p* < 0.001) and right adhesion strength (*r* = 0.227, *p* < 0.001). In contrast, the Vt showed a negative correlation with the bottom adhesion strength (*r* = −0.057, *p* = 0.025) (Table 3). The correlation coefficient size had a relationship with Vt in the order PP, Left, Right, ln(apex), and ln(bottom).

### 3.3. Factors Influencing Vt

The multiple regression analysis results were significant (F = 302.865, *p* < 0.001) with an explanatory value of 64.3%. Both the PP (β = 0.797, *p* < 0.001) and bottom (β = −0.186, *p* < 0.001), left (β = 0.172, *p* < 0.001) and right adhesion strengths (β = −0.123, *p* < 0.001) were identified as factors affecting the Vt (Table 4). When the sizes of the standardization coefficients were compared, it was confirmed that bottom, left, and right had a greater effect on Vt in that order.

## 4. Discussion

This study aimed at identifying factors that affected the Vt and determining the most appropriate location of the hand between the mask and face to provide an optimal Vt when bagging the BVM ventilator using the one-handed EC grip. 

The study showed that most participants (92.6%) used their left hand to hold the mask, regardless of hand dominance, which was similar to the observations made in a previous study [20]. The number of ventilations measured was 1–33 within 2 min depending on the participant. In some participants, the number of ventilations performed was ≤ 20 because the mask did not seal well to the manikin’s face; consequently, it was assumed that air leakage occurred. However, from more than 24 measurements it was observed that bagging was too quick, which may have eventually caused side effects by providing excessive amounts of air to the patients. Therefore, training on using a metronome is necessary to provide accurate ventilation. 

Although the average Vt measured in this study (415.98 mL) was similar to that obtained by Lee et al. [21], the volume recommended by the American Heart Association (6−7 mL/kg) was not achieved. At the average weight of 71.2 kg of a South Korean adult [22], BVM ventilation should provide a Vt of 427.2–498.4 mL. 

The PP is the maximum pressure applied to the lungs during one round of ventilation, and a previous study reported that the maximum allowable range of PP that does not cause adverse effects is ≤40 cmH_2_O [21]. In this study, the average PP was 15.56 cmH_2_O, which was higher than the pressure applied by Uhm and Kim (11.62 cmH_2_O) [21] and Shin et al. (8.71 cmH_2_O) [18]. Based on the method used in a previous study [23], the PP that was applied in this study was within the optimal level range for the prevention of adverse effects. 

Among the 4-spot adhesion strengths, the force at the bottom contact point was lower than that measured at the right and left of the mask. However, the bottom sealing force was the highest and that the apex sealing force was the weakest, which was in line with the results of Uhm and Kim [24]. Given the paucity of investigations that analyzed the 4-spot adhesion strength between the mask and face, it was difficult to explain the disparities between the findings. However, they were likely due to differences in the study designs, experimental environments, and participants’ physical conditions. 

In the correlation analysis, all the experimental variables for the Vt were significant. Consequently, the Vt increased as the PP, apex, and left and right adhesion strengths increased. However, after adjusting for the general characteristics and PP variables, the multiple regression analysis confirmed that the Vt increased when the bottom and right sealing pressure was lower, and the left sealing pressure higher. This was the result of controlling for gender, grade, and orientation of the hand that sealed the mask. This result is in the same context as a study that emphasized a higher sealing force between the thumb and index finger compared to the existing EC technique [11] and a study that the ventilation increased as the distance between the thumb and index finger increased [20,24]. However, to accurately identify the cause of these results, follow-up studies are needed.

Therefore, first, the force of the second finger should be reduced to lower the sealing force in the lower right corner. Umesh et al. [20] reported that in order to lower the bottom pressure, it is necessary to release the force of the index finger that makes the bottom of the “C” shape around the mask. Second, to increase the sealing pressure on the left and to decrease the sealing pressure on the right, it may help to adjust the pressures by applying pressure to the curved area between the thumb and index finger of the left hand by holding the mask in close contact with the left hand. Furthermore, the airway openness must be maintained and not overlooked.

Joffe et al. reported that “a two-handed jaw-thrust technique” increased the tidal volume by securing the upper airway better than the EC technique [25]. However, in South Korea, when considering the manpower put into the field, it is difficult for two rescuers to be put into ventilation. So based on the results of this study, one rescuer should implement evidence-based ventilation.

In conclusion, this study is meaningful in suggesting that it is necessary to maintain the E technique, which opens the airway while applying force to the left side of the flexion of the left hand, in order to effectively seal the mask despite the limitations.

### Limitations

This study had several limitations. First, it was difficult to generalize the results of a manikin study to actual clinical scenarios. These results should be interpreted with caution because the general characteristics and PP were used as control variables without taking into account the physical characteristics related to the participants’ hands. Second, the results of this study may not be generalizable because the participants were not fully representative of undergraduate paramedic student population in South Korea. Finally, this study design does not include explanations of causal relationships with latent variables that may exist. Therefore, further studies are needed to determine the causal relationship in detail.

## 5. Conclusions

Based on the findings of this study, it is necessary to bottom seal the mask with the space between the thumb and index finger while holding the mask with the left hand and releasing the force on the second finger that seals the floor. There should be greater force in the left flank of the “C” shape (the left contact spot of the mask) than in the other three places (apex, bottom, and right contact spots of the mask) for optimal Vt with collapsed patients, when using the one-handed EC technique. Additionally, for the provision of accurate ventilation it may be necessary to provide training using a metronome. The findings of this study, therefore, present a practical method to provide the optimal Vt to patients who require ventilation. Further research is required to ensure the reliability and validity of the results of this study. 

## Figures and Tables

**Figure 1 healthcare-10-01365-f001:**
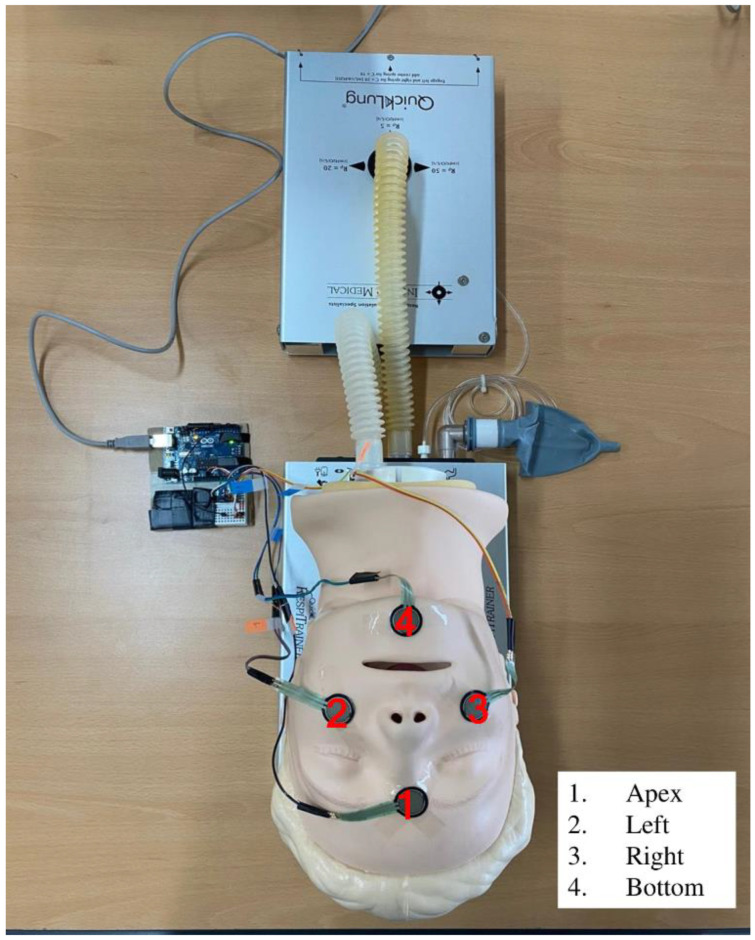
The four sensor attachment sites.

**Table 1 healthcare-10-01365-t001:** General characteristics, *N* = 89.

		*N* (%)
Sex	Male	41 (46.1)
	Female	48 (53.9)
School year	Sophomore	28 (31.5)
	Junior	31 (34.8)
	Senior	30 (33.7)
Hand holding the mask	Left	82 (92.1)
	Right	7 (7.9)

Bag valve mask ventilation was performed 1–33 times by each participant for 2 min, and 1524 data points were collected and analyzed.

**Table 2 healthcare-10-01365-t002:** The mean and range of the study main variables, N = 1524.

			N	(%)	Mean (±SD)	Measuring Range
General characteristics as the control variables	Sex	Male	764	(50.1)		
		Female	760	(49.9)		
	School year	Sophomore	519	(34.1)		
		Junior	552	(36.2)		
		Senior	453	(29.7)		
	Hand holding the mask	Left	1411	(92.6)		
		Right	113	(7.4)		
	PP (cmH_2_O)				15.56 (±5.62)	5–200 cmH_2_O/L/s
Experimental variables	4-spot adhesion strength -ln(apex) (N)				0.03 (±0.07)	0–100 N
	4-spot adhesion strength- ln(bottom) (N)				0.17 (±0.29)	0–100 N
	4-spot adhesion strength- left (N)				0.62 (±0.49)	0–100 N
	4-spot adhesion strength- right (N)				0.69 (±0.63)	0–100 N
Dependent variable	Tidal Volume (mL)				415.98 (±102.4)	0–1200 mL

PP: peak pressure; ln(apex) = ln(apex + 1)^6^, ln(bottom) = ln(bottom + 1).

**Table 3 healthcare-10-01365-t003:** Correlation between the tidal volume, peek pressure, and 4-spot adhesion strengths.

	*r*	(*p*)
Tidal Volume	1	
PP	0.744 ***	(<0.001)
ln(apex)	0.083 **	(0.001)
ln(bottom)	−0.057 *	(0.025)
Left	0.280 ***	(<0.001)
Right	0.227 ***	(<0.001)

PP: peak pressure; ln: ln(apex) = ln(apex+1)^6^, ln(bottom) = ln(bottom+1); * *p* < 0.05, ** *p* < 0.01, *** *p* < 0.001.

**Table 4 healthcare-10-01365-t004:** Factors influencing the tidal volume.

	B	S.E.	β	t	*p*
(constant)	199.804	6.053		33.011 ***	<0.001
ln(apex)	−32.060	25.522	−0.021	−1.256	0.209
ln(bottom)	−65.440	5.781	−0.186	−11.320 ***	<0.001
Left	35.766	3.576	0.172	10.002 ***	<0.001
Right	−20.156	3.281	−0.123	−6.144 ***	<0.001
PP	14.519	0.315	0.797	46.084 ***	<0.001
Sex (male = ref.)					
Female	−27.241	3.430	−0.133	−7.942 ***	<0.001
School year (sophomore = ref.)					
Junior	22.574	3.918	0.106	5.761 ***	<0.001
Senior	−1.082	4.387	−0.005	−0.247	0.805
Hand holding the mask (left = ref.)					
Right	−4.643	6.205	−0.012	−0.748	0.454
	F = 302.865 ***, R^2^(_adj_R^2^) = 0.643 (0.641)				

PP: peak pressure; ln: In(apex)=ln(apex + 1)^6^, ln(bottom) = ln(bottom + 1); ref. = reference variable; *** *p* < 0.001.

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
