# Peer review of "Potential Maneuvers for Providing Optimal Tidal Volume Using the One-Handed EC Technique"

_healthcare, 2022, doi:10.3390/healthcare10081365_

Round 1
Reviewer 1 Report
This manuscript is developed in the framework of bag-valve-masks (BVM), a device that is commonly used in patients with respiratory failure. The adequate BVM ventilation is highly related with survival rates, a reason for maintaining optimal ventilation. A parameter which is related with the quality of respiration is the tidal volume. In this paper, the authors performed a study which was aimed to identify which factors influence the tidal volume.
After a detailed review, I consider that the topic might result of interest. However, I have some doubts and questions. For that reason I recommend to perform a Major Revision.
Please see my comments in the attached pdf file.

Author Response
Please see the attachment.
Thank you for your meticulous review.

Reviewer 2 Report
The aim of the article is to determine the most suitable location of the hand between the mask and face for providing optimal Vt while using BVM ventilation with the one-handed EC technique. The investigation is very important for patients with lung diseases.
The article is written clearly. The only my remark is related to the lack of causality analysis. I suggest to analyze the causes of the results more in detail.
Author Response

(The authors gave the same response as above.)

Round 2
Reviewer 1 Report
The authors have replied to all my questions.
I think that it can be now considered for publication.